# The Invisible Fraction within Melanin Capable of Absorbing UV Light and with Fluorescent Properties: Is It Lacking Consideration?

**DOI:** 10.3390/ijms25158490

**Published:** 2024-08-03

**Authors:** Aaliyah Flake, Koen Vercruysse

**Affiliations:** 1College of Medicine, UTHSC, Memphis, TN 38163, USA; aflake1@uthsc.edu; 2Chemistry Department, Tennessee State University, Nashville, TN 37209, USA

**Keywords:** melanin, catecholamines, catechols, serotonin, UV-Vis spectroscopy, FT-IR spectroscopy

## Abstract

Expanding on earlier observations, we show that many melanin materials, in vitro synthesized from a wide range of precursors, can be fractionated into a dark-colored precipitate and a near-colorless, dispersible fraction. The dispersible fractions exhibited absorbance in the UVA and UVB range of the electromagnetic spectrum, but none in the visible range. In addition, fluorescent properties were associated with all dispersible fractions obtained. FT-IR spectroscopic analyses were performed to compare both types of fractions. Overall, it appears that some of the properties associated with melanin (UV absorbance, fluorescence) may not necessarily reside in the dark-colored portion of melanin, but in a colorless fraction of the material. It remains to be seen whether any of these in vitro observations have any relevance in vivo. However, we raise the possibility that the presence of a colorless fraction within melanin materials and their associated properties may have received inadequate attention. Given the important association between melanin, UV protection, and skin cancer, it is worthwhile to consider this additional aspect of melanin chemistry.

## 1. Introduction

Melanin (MN), a ubiquitous class of darkly colored pigments, is poorly defined in terms of its chemical structure and some of its physical properties. Many reviews have been written about this enigmatic class of biomolecules detailing these unresolved issues [1,2,3,4,5,6,7,8]. There is consensus that MN is formed through the oxidation of phenolic precursors, and a wide variety of such precursors can generate MN-like materials. In human and other animal species’ physiology, two distinct classes of MNs are responsible for the coloration of skin and hair: eumelanin (EuMN) and pheomelanin (PhMN) [9,10]. EuMN is typically described as a brown- to black-colored material built from L-DOPA as the precursor. PhMN is typically described as a yellow to red pigment formed by a combination of L-DOPA and the amino acid cysteine. The biochemical pathways that lead to these two different classes of MN are described by the classic Raper–Mason scheme [3,11,12,13]. MN is a protective entity against sunlight-induced damage due to its capacity to absorb light over a broad range of the electromagnetic spectrum, including the UV portion. However, this protection depends on the type of MN, EuMN or PhMN, present or the ratio at which these two are present. The presence of elevated levels of PhMN in an individual’s skin is a risk factor making the individual more prone to sunlight-induced skin damage, potentially leading to an increased risk of skin cancer, like cutaneous melanoma (CM) [14,15]. CM is an aggressive form of skin cancer that has been studied extensively. The impact of MN, either as EuMN or PhMN, on CM’s initiation, morphology, progression, metastasis, or recurrence has been discussed and reviewed [16,17,18]. Overall, the effect of MN, beneficial or detrimental, is uncertain. This uncertainty could in part be attributed to the uncertainties surrounding the chemical structure of MN. MN is not a single molecule but is typically considered a combination of chemically related entities [10,19]. In addition, MN’s chemistry, biosynthesis, and secretion in vivo are heavily regulated and not constant [16]. The best insights into the chemical structures of the components of MN came from mass-spectroscopic analyses. However, many uncertainties remain because the various reported studies employed different reaction conditions for the synthesis of MN materials. Some studies involved the use of enzymatic [20,21,22,23,24,25,26] or non-enzymatic approaches [22,27]. Different precursors, like dopamine [21,22,26,28,29,30,31,32], DOPA [20,22,26,27], DHI [22,24,25,28,29], or DHICA [22,23], were used. Different oxidizing agents, like O_2_ [20,21,22,23,25,26,27,28,29,30,31,32], H_2_O_2_ [22,23,24,25], NaIO_4_ [23,25,32], or K_3_Fe(CN)_6_ [23,25], were used. Some studies focused on the analysis of the dispersible portion (supernatant or filtrate) of the reaction mixture [20,21,22,26,27,30] while others focused on the analysis of the precipitated material (e.g., coatings) [23,24,25,28,29,30,31]. Some studies focused on the initial stages of the reaction (first few minutes to hours) [20,21,22,23,24,25,26,27,29,30] while others focused on the later stages of the reaction (24 h or more) [24,28,29,30]. Thus, a wide range of reaction products have been observed and reported upon: (1) oligomeric [20,21,22,23,24,25,26,27,29,30,32], polymeric [31], or mixed oligomeric and polymeric species [22], (2) species that have undergone additional oxidative reactions [21,22,23,25,26,29], (3) species with additional cyclic entities [28,30], (4) species formed by opening of the catecholic ring structures [23,25], or (5) non-covalent aggregates [27,30,32]. Due to its insolubility in neither water nor organic solvents, typical chemical analysis techniques are not readily applied to the study of MN. UV-Vis spectra of MN materials, stably dispersed in water, and FT-IR spectra of solid MN materials are frequently reported. UV-Vis spectra will typically show a monotonic profile over the entire UV-Vis range, with some absorbance bands in the UV range [7]. The interpretation of the FT-IR profile of any MN-like material is difficult as the signals are broad, indicative of its heterogeneous and/or polymeric nature. In addition, the multitude of functional groups and types of bonds present will lead to a multitude of signals that often overlap. A summary of typical FT-IR assignments associated with MN materials is presented by Pralea et al. [7] Given the association of MN with dark coloration, the term “invisible melanin” seems a contradiction. However, it stems from earlier observations that the in vitro synthesis of MN leads to a colloidal, hybrid material consisting of a dark-colored, EuMN-like substance, and a colorless substance [33,34]. A model has been proposed in which the colorless substances serve as stabilizing ligands keeping the EuMN aggregates in suspension. Altering the ionic strength of the mixture through the addition of cationic species leads to a precipitation of the EuMN material and a dissociation of the invisible substance from the EuMN aggregates [34]. The possibility that an aspect of the chemistry behind the synthesis of MN has been overlooked was raised recently [35]. In that research, two chemically and physically distinct materials were generated from dopamine as the precursor. This report expands our earlier observations by expanding the pool of precursors used to synthesize MN-like materials. Figure 1 presents the chemical structures of all precursors used in this study.

From each precursor, MN-like materials were generated through air-oxidation in the presence of Na_2_CO_3_. Whenever possible the MN materials were fractionated into a dark substance and a near-colorless substance through the addition of LaCl_3_. All fractions thus obtained were characterized and compared through spectroscopic analyses. The results and discussions presented here are solely based upon in vitro observations. Any possible relevancies to in vivo situations are to be considered but are mere hypotheses.

## 2. Results

### 2.1. Observations during Synthesis of Materials

#### 2.1.1. DHI

Although synthetic DHI is described as a light gray or off-white powder [22,29], the commercial DHI material we received and used in the experiments reported herein consisted of a dark powder as shown in Appendix A. When dispersed in water, a mixture with a dark purple appearance was obtained. Appendix A, shows an absorbance spectrum (visible region) of a sample containing 3 mM DHI dispersed in water. The spectrum shows a monotonic profile, similar to that of a MN-like material but with a broad absorbance band centered around 550 nm superimposed on it. Appendix A shows the FT-IR spectrum of the commercial DHI prior to any reaction. The FT-IR spectrum features a multitude of sharp absorbance peaks similar to that of a singular compound. It does not feature the broad absorbance bands of a typical MN material. These observations regarding the starting DHI material do suggest that some of the DHI material had already undergone some potential oxidation reaction. When this DHI was reacted in the presence of Na_2_CO_3_, the mixture darkened further very quickly. No precipitations were present in the mixture at any time during the reaction. However, upon mixing aliquots of the reaction mixture with solvent for HPLC analysis, dark precipitations formed, and the supernatant appeared to be almost colorless. HPLC traces of the supernatant of these aliquots would typically show signals associated with unreacted DHI and small signals associated with reaction products.

#### 2.1.2. Serotonin

The reaction involving serotonin produced much precipitation and much coating of the glass reaction container and the stirring bar used. Appendix A shows photographs of the coated glassware and stirring bar. Serotonin is known to readily produce films when oxidized under alkaline conditions [36]. HPLC analyses of the supernatant of the reaction mixture during the reaction revealed a major signal associated with unreacted serotonin and little to no other signals.

#### 2.1.3. Other Precursors

All other reaction mixtures presented themselves as homogenous mixtures with no visible signs of precipitation. For all these other mixtures, absorbance spectra could be recorded following dilution with water. Appendix A presents k values obtained through exponential regression of the spectra of the crude reaction mixtures and their AUC values calculated according to Equation (1) (see Section 4.5). While the value of the AUC depends on the concentration of the sample, the value of k does not and is a characteristic of the material synthesized. Appendix A shows a plot of the A_650_/A_500_ values vs. k for all the crude reaction mixtures in relationship to the theoretical line associated with Equation (2) (see Section 4.5). These results show that Equation (1) is applicable to the visible-range absorption spectra of all the MN materials synthesized as all the data points fall on or near the theoretical line. The critical difference among all the materials is their value of k, which is associated with the appearance of the material: dark- or light-colored [37]. E.g., because of its relatively high AUC value and relatively low k value, the reaction mixture involving DHI presented itself as a black mixture. In contrast, the mixture involving pyrogallol presented itself as a dark yellow mixture due to its relatively high value of AUC and relatively high value of k. Thus, based upon their appearances, both EuMN-like and PhMN-like materials were generated. PhMN is typically described as a yellow-to-red material made in the presence of cysteine. This may only be applicable to in vivo synthesis of PhMN. Our lab has shown that in vitro, the addition of cysteine to the reaction mixtures leads to darker, more EuMN-like materials [37]. On the other hand, yellow-to-red materials can be generated if one uses precursors like pyrogallol, epinephrine, or others as indicated in this report. In this context, it is difficult (and possibly confusing) to define EuMN- or PhMN-like for in vitro synthesized materials. One can define PhMN as a melanin material synthesized in the presence of cysteine or as a melanin material with a yellow-to-red appearance, but in vitro these two aspects do not merge as our previous research has shown [37].

### 2.2. La^3+^ Precipitation Tests of Crude Reaction Mixtures

Except for the reaction involving serotonin, a small-scale precipitation test involving La^3+^ was performed on the dialyzed crude reaction mixtures as outlined in Section 4.3. AUC values calculated according to Equation (1) of the resulting supernatants were plotted as a function of the La^3+^ concentration present. Figure 2 presents the results thus obtained. For clarity purposes, the results were grouped into two panels: L-DOPA, dopamine, norepinephrine, epinephrine, and DHI in panel (a), and catechol, pyrogallol, 3,4-dihydroxybenzoic acid, and caffeic acid in panel (b).

In the case of epinephrine, little to no changes in AUC values could be observed, indicative of the fact that no precipitation occurred in the presence of La^3+^. The results obtained are comparable to similar experimental results obtained for the case of L-DOPA [34].

### 2.3. Fractionation of MN Materials

The results of the small-scale tests described in Section 4.2 allowed for an estimate of the minimum amount of La^3+^ needed to precipitate the dark-colored material and obtain colorless or minimally colored supernatants from the dialyzed reaction mixtures as outlined in Section 4.3. For most mixtures, a dark-colored, precipitated fraction (termed F_prec_) and a dispersible, colorless to light-colored fraction (termed F_disp_) could be obtained as described in Section 4.3. Exceptions were epinephrine, for which only a F_disp_ fraction was obtained, and DHI and serotonin, for which only F_prec_ fractions were obtained. Appendix A shows photographs of the F_disp_ and F_prec_ fractions obtained from the various precursors. Appendix A includes data on the amounts of materials thus obtained, keeping in mind that the materials probably exist as complexes with La^3+^.

### 2.4. Characterization of Fractions

#### 2.4.1. UV-Vis Spectra of F_disp_ Fractions

Figure 3 shows the UV-Vis spectra of the various F_disp_ fractions dispersed in water at 100 μg/mL. For clarity purposes, the results were grouped into two panels: L-DOPA, dopamine, norepinephrine, and epinephrine in panel (a), and catechol, pyrogallol, 3,4-dihydroxybenzoic acid, and caffeic acid in panel (b).

All UV-Vis profiles of the F_disp_ fractions display absorbance bands around 280 nm and between 300 and 400 nm, covering the UVA (315–400 nm) and UVB (280–315 nm) range of the electromagnetic spectrum. The profiles of the F_disp_ fractions show little to no absorbance above 400 to 450 nm. This correlates with their light-yellow to colorless appearances.

#### 2.4.2. Concentration-Dependent Fluorescence of F_disp_ Fractions

Figure 4 shows the concentration-dependent fluorescence recorded for the various F_disp_ fractions. For clarity purposes, the results were grouped into two panels: L-DOPA, dopamine, norepinephrine, and epinephrine in panel (a), and catechol, pyrogallol, 3,4-dihydroxybenzoic acid, and caffeic acid in panel (b).

All F_disp_ fractions showed concentration-dependent fluorescent properties, although with differences in intensity depending on the precursor involved. The results are in line with earlier observations [38,39].

#### 2.4.3. FT-IR Spectroscopy

Figure 5, panels a through j, show the FT-IR spectra of F_disp_, F_prec_ before and F_prec_ after washing with 1 N HCl. Each panel is focused on the materials obtained from a single precursor, and all spectra are normalized for their absorbance at 1600 cm^−1^ for comparison purposes.

Overall, the FT-IR spectra exhibit the typical features of MN-like materials. The FT-IR spectra of the F_disp_ and F_prec_ fractions show similar patterns in absorbance bands. However, there are clear qualitative differences between wavenumbers 1000 and 1500 cm^−1^ hinting at chemical differences between both fractions, as observed before [33,34]. The FT-IR spectra obtained from F_prec_ washed with 1 N HCl yielded a clear signal (distinct peak or shoulder) in the low 1700 cm^−1^ range. An exception to this was the case of serotonin (see Figure 5j). The appearance of a stronger signal in the low 1700 cm^−1^ range upon washing of F_prec_ with HCl suggests the presence of carboxylic acid functional groups in these F_prec_ materials (with the exception of serotonin). Carboxylic acid functions can be expected in many MN materials due to the presence of this functional group in the precursor (see Figure 1) or due to catechol ring opening leading to the emergence of carboxylic acids [23,25]. In this context it is important to distinguish the FT-IR spectroscopic features of carboxylic acids vs. carboxylates. A typical FT-IR feature of carboxylic acids is a signal between 1700 and 1730 cm^−1^ due to the presence of the C=O group. This signal can appear between 1640 and 1700 cm^−1^ for C=O groups conjugated to unsaturated bonds [40]. Such a signal is not to be observed in the case of carboxylate functionalities. Carboxylates exhibit two signals related to the asymmetric (between 1540 and 1650 cm^−1^) and symmetric C-O stretches (between 1360 and 1450 cm^−1^) [41]. These aspects can be illustrated by the FT-IR spectra of caffeic acid, 3,4-dihydroxybenzoic acid, and L-DOPA, as shown in Appendix A. Both caffeic acid and 3,4-dihydroxybenzoic acid contain a conjugated carboxylic acid functionality (see Figure 1) and exhibit a distinct signal in their FT-IR spectra at 1641 cm^−1^ and 1668 cm^−1^ respectively. L-DOPA, as shown in Figure 1, does contain a non-conjugated, carboxylic acid functionality. However, its FT-IR spectrum does not exhibit a signal between 1700 and 1730 cm^−1^. This suggests that L-DOPA is present in its zwitterion form and contains a carboxylate functional group. The FT-IR spectrum of L-DOPA does show distinct signals between 1560 and 1650 cm^−1^ and between 1350 and 1440 cm^−1^ that can be attributed to the presence of carboxylate. The existence as a zwitterion implies the presence of a primary amine salt (-NH_3_^+^) in the structure of L-DOPA. Primary amine salts exhibit signals between 2800 and 3200 cm^−1^ (N-H stretch) and two signals: one between 1560 and 1625 cm^−1^ and one between 1500 and 1550 cm^−1^ associated with N-H bend [42]. The FT-IR spectrum of L-DOPA does show an array of signals between 1500 and 1600 cm^−1^, which can be attributed to the presence of a primary amine salt in addition to the signals associated with the presence of a carboxylate group. If carboxylic acid functional groups are present in the MN material, it is reasonable to assume that, upon fractionation with La^3+^, they would exist in carboxylate form. Hence, most of the materials obtained through fractionation with La^3+^ exhibit no signal in the low 1700 cm^−1^ range but do exhibit a double signal between 1540 and 1650 cm^−1^, and between 1360 and 1450 cm^−1^. The washing with HCl converts these carboxylates into carboxylic acids, and a distinct signal in the low 1700 cm^−1^ range emerges.

## 3. Discussion

The results presented in this report confirm and expand on our earlier observations that in vitro synthesized MN can be fractionated into a dark-colored precipitate and a colorless, dispersible fraction [33,34]. Both fractions are synthesized simultaneously during the reaction and are generated from the same precursor. Thus, both fractions could be considered as additives to each other. The fact that a simple fractionation process using multivalent cations can separate the colorless fraction from the colored fraction indicates that a non-covalent interaction was formed between the two during their synthesis. A previous study used Ca^2+^ to achieve fractionation [33]. Subsequent studies indicated that trivalent lanthanide cations yielded better separations [34]. Thus, La^3+^ was used in this study to fractionate the MN materials as it does not interfere with UV-Vis or fluorescence analyses. With few exceptions, MN materials generated from a variety of precursors could readily be fractionated following the dialysis of the reaction mixture. An exception was the case of epinephrine, which yielded only a F_dsip_ fraction and no EuMN-like F_prec_ fraction. Although epinephrine by itself appears not to generate any EuMN-like materials, in the presence of cysteine or other amino acids it does [37]. Other exceptions were DHI and serotonin which did not yield any F_disp_ fraction. Although DHI and serotonin appear not to generate any colorless fraction, this does not exclude the possibility that colorless substances are embedded within the dark substances and might need a different fractionation approach. Many parameters affect the synthesis of MN, but ionic strength is often not controlled nor discussed [7,43]. However, our current and previous observations do suggest that, in many cases, ionic strength affects the physical stability of the MN materials synthesized. Although MNs are described as insoluble in water or organic solvents, the in vitro synthesis of MN often leads to seemingly dissolved entities. The reality is that MNs often present themselves as physically stabilized, colloidal particles. In an earlier report focusing on L-DOPA, we presented evidence that some colorless reaction products may serve as stabilizing ligands for the dark-colored MN particles [34]. These earlier observations have been expanded in this report by expanding the pool of precursors that can lead to MN-like materials. The physical stability of a colloid depends on many factors as reviewed elsewhere [44]. Regarding the stability of MN colloids, apart from the synthesis conditions, the type of precursor may play an important role. This is evidenced in the vastly different results obtained when using serotonin or epinephrine as the precursor. We suspect that the physical stability of the colloidal MN materials depends, in part, on their anionic surface properties due to the presence of carboxylic acid functionalities. The presence of such functionalities can be judged from FT-IR spectroscopic analyses, as discussed earlier. Carboxylic acid or carboxylate functional groups are to be expected in MN materials synthesized from L-DOPA, 3,4-dihydroxybenzoic acid, or caffeic acid. However, the observation of the presence of carboxylic acids in the MN materials made from dopamine, norepinephrine, catechol, pyrogallol, and DHI suggests that catecholic ring opening occurred during the synthesis of these MN materials. The apparent absence of carboxylic acid functionalities in the MN material synthesized from serotonin may explain its physical instability and its tendency to coat surfaces, as shown in Appendix A. On the other hand, serotonin-based MN materials can be expected to contain primary amine functional groups, which would be protonated in the presence of HCl. This might explain the dispersion of the F_prec_ fraction obtained from serotonin in HCl solution. The uncharged, anionic, cationic, or zwitterion character of any MN material may impact their in vitro properties and applications. In vivo, the surface properties of MN materials may impact their binding and interaction with proteins or other biomolecules and their tendency to precipitate or not. In this regard, and without making any further inferences, it is important to point out that serotonin is synthesized and present in skin cells, including melanocytes [45]. In addition, the cells in the digestive system that serve as the major source of serotonin have been reported as being darkly colored [46]. The UV-Vis spectra of the F_disp_ fractions show strong absorbance within the UV region of the electromagnetic spectrum with little to no absorbance in the visible region. Hence, their description as “invisible melanin”. Thus, the question raises itself whether the UV absorbance typically ascribed to MN materials resides in this “invisible”, F_disp_ fraction and not in the dark, F_prec_, fraction. Similarly, the fluorescent properties of MN materials, which have been a topic of debate [47,48], may reside in the colorless fraction of the material, while the dark fraction would be responsible for the quenching of this fluorescence. The F_prec_ fraction is associated with the dark color typical of MN-like materials. This dominance of the physical appearance of the MN material may have tilted the research on MN materials away from the light-colored or colorless fractions that may be present. For any physico-chemical property ascribed to MN, the question would have to be asked if these properties are associated with the F_disp_ or the F_prec_ fraction or exist because of a synergistic action of both fractions. Our observations show that the absorbance features of the colorless F_disp_ materials make them ideal absorbers of UVA and UVB radiation. This implies that the dark component associated with MN may not be necessary to absorb UV light. However, it may be involved in the dispersion of the absorbed energy. Thus, the following questions could be asked: (a) is a similar type of “invisible melanin” generated during the in vivo melanogenesis process, and (b) if so, is this type of “invisible melanin” ultimately responsible for the UV absorbance in vivo? It is worth noting that in a study of neuromelanin of the human brain, soluble and insoluble components were described [49]. The possibility of an overlooked aspect of the chemistry behind MN goes beyond the issue of CM. It has been established that a relationship exists between CM and Parkinson’s disease (PD) [50], and MN has been implicated in the physiology of PD and other degenerative diseases, like Alzheimer’s disease [51].

## 4. Materials and Methods

### 4.1. Materials and Solutions

Dopamine.HCl (Thermo Scientific, Waltham, MA, USA), L-DOPA (MP Biomedicals, Santa Ana, CA, USA), epinephrine (Alfa Aesar, Ward Hill, MA, USA), norepinephrine.HCl (Sigma, Tokyo, Japan), catechol (Acros Organics, Geel, Belgium), pyrogallol (Sigma), caffeic acid (Sigma), 3,4-dihydroxybenzoic acid (Aldrich, St. Louis, MO, USA), serotonin.HCl (Thermo Scientific), 5,6 dihydroxyindole (DHI; Thermo Scientific), and LaCl_3_·7H_2_O (Thermo Scientific) were all purchased through Fisher Scientific (Waltham, MA, USA) but originated from different suppliers as indicated.

### 4.2. Synthesis of MN Materials

In a volume of 100 mL, 250 mg precursor was dissolved in 25 mM (catechol, pyrogallol, DHI, L-DOPA) or 50 mM Na_2_CO_3_ (dopamine.HCl, norepinephrine.HCl, caffeic acid, 3,4-dihydroxybenzoic acid, serotonin.HCl). The higher concentration of Na_2_CO_3_ was used for the precursors in hydrochloride form or for the precursors that contain a carboxylic acid functionality (the exception being L-DOPA, which exists in zwitterion form). In the case of epinephrine, 250 mg was dispersed in 50 mL water, and 100 μL glacial acetic acid was added to dissolve the compound. The reaction was initiated by the addition of 100 mL of 50 mM Na_2_CO_3_. Reaction mixtures were kept stirring at room temperature until the precursor had reacted away as judged from RP-HPLC analyses (between two and seven days, depending on the precursor). An exception was the reaction involving serotonin. Serotonin did not completely react after ten days. This reaction mixture yielded dark precipitates and very few substances in the supernatant portion of the mixture. After ten days of reaction, this mixture was centrifuged, and the precipitate was washed extensively with water and lyophilized. All other reaction mixtures were dialyzed against 3.5 L water for two days with four changes of water.

### 4.3. Fractionation of Dialyzed MN Reaction Mixtures

Following dialysis, 500 μL aliquots from the reaction mixtures were mixed with 50 μL water or 50 μL La^3+^ solutions at varying concentrations such that La^3+^ concentrations ranged from 0 to 5 mM. After three hours standing at room temperature, the mixtures were centrifuged, and an absorbance spectrum of the supernatant was recorded. These spectra were used to determine the minimum amount of La^3+^ needed to obtain maximum precipitation of the dark-colored material generated within the crude reaction mixture. Based upon the results thus obtained, 10 mL La^3+^ solution between 25 and 50 mM, depending on the precursor involved, was added to the remaining dialyzed reaction mixture. After standing overnight at room temperature, the mixture was centrifuged, and the supernatant lyophilized without any further processing. This fraction was termed the dispersible fraction (F_disp_). The precipitate was washed repeatedly with water and lyophilized. This fraction was termed the precipitated fraction (F_prec_).

### 4.4. Washing of Materials with HCl

Lyophilized F_disp_ and F_prec_ materials were dispersed in 1 N HCl to a concentration of 25 mg/mL. The F_disp_ materials yielded stable dispersions when mixed with 1 N HCl and no further analyses were performed. When mixed with 1 N HCl, the F_prec_ material obtained from 3,4-dihydroxybenzoic acid yielded only a physically stable dispersion, and no further analyses were performed. When mixed with 1 N HCl, the precipitate obtained from serotonin yielded a physically stable dispersion. This mixture was diluted 10-fold with water and lyophilized. When mixed with 1 N HCl, F_prec_ materials made from catechol and caffeic acid yielded both a physically stable dispersion and a precipitate. The supernatants were not characterized further, while the precipitates were washed with water and lyophilized. All other F_prec_ materials yielded a precipitate only when mixed with 1 N HCl. These precipitates were washed with water and lyophilized.

### 4.5. UV-Vis Spectroscopy

Lyophilized F_disp_ fractions were dispersed in water at a concentration of 1 mg/mL, and dilutions were prepared from this stock solution. UV-Vis spectroscopic measurements were made in wells of a 96-well microplate using a SynergyHT microplate reader from Biotek (Winooski, VT, USA). For measurements involving absorbance readings below 350 nm, UV-transparent microplates were used. Quantitative estimates of the “darkness” of any sample were calculated by integrating, between 400 and 900 nm, the absorbance spectrum of the sample fitted with an exponential function and calculating the area-under-the-curve (AUC), as outlined elsewhere and shown in Equation (1) [52].
(1)$$AUC=\int_{400}^{900}A_{0}*e^{-k\lambda }d\lambda =\frac{A_{0}}{-k}*\left(e^{-k*900}-e^{-k*400}\right)$$

In this equation, k is the decay constant of the exponential profile of the absorbance (A) as a function of the wavelength (λ). In addition, to distinguish EuMN-like materials from PhMN-like materials, the ratio of the absorbance at 650 nm over the absorbance at 500 nm (A_650_/A_500_) was evaluated. Although these two wavelengths are arbitrarily chosen, it does follow precedent as a suggested way to differentiate dark-colored EuMN from light-colored PhMN [5,37,53,54]. Given the exponential relationship between A and λ, an exponential relationship exists between k and A_650_/A_500_ as shown in Equation (2).
(2)$$\frac{A_{650}}{A_{500}}=\frac{A_{0}*\;e^{-k*650}}{A_{0}*e^{-k*500}}=\frac{e^{-k*650}}{e^{-k*500}}=e^{-k*\left(650-500\right)}=e^{-k*150}$$

### 4.6. Fluorescence Spectroscopy

Lyophilized F_disp_ fractions were dispersed in water at a concentration of 1 mg/mL, and dilutions were prepared from this stock solution. Fluorescence measurements were made in wells of an opaque 96-well microplate using a SynergyHT microplate reader from Biotek (Winooski, VT, USA) with the excitation filter set at 360 nm, emission set at 460 nm, and the sensitivity factor set at 75.

### 4.7. High-Performance Liquid Chromatography (HPLC)

HPLC analyses were performed on a Breeze 2 HPLC system equipped with a 1500 series HPLC pump and a model 2998 photodiode array detector from Waters, Co. (Milford, MA, USA). Analyses were performed using a Synergi C8 column (Phenomenex, Torrance, CA, USA) in isocratic fashion at a flow rate of 0.5 mL/min using a mixture of 25 mM Na-acetate:methanol:acetic acid (90:10:0.05% *v*/*v*) as solvent. The pH of this solvent was measured to be 5.3.

### 4.8. FT-IR Spectroscopy

FT-IR spectra of all lyophilized materials were recorded. FT-IR analyses were performed using a Spectrum Two FT-IR spectrometer from PerkinElmer (Waltham, MA, USA). Scans were made using the universal ATR accessory between 650 and 4000 cm^−1^ with a resolution of 4 cm^−1^ and using the OptKBr beam splitter and LiTaO_3_ detector. For each sample, 24 scans were accumulated.

### 4.9. Dialysis and Freeze Drying

Crude reaction mixtures were dialyzed using Spectrum Spectra/Por RC dialysis membranes with a molecular-weight-cut-off (MWCO) of 3.5 kDa, obtained from Fisher Scientific (Waltham, MA, USA). Freeze drying was performed using a Labconco FreeZone Plus 4.5 L benchtop freeze-dry system obtained from Fisher Scientific (Waltham, MA, USA).

## 5. Conclusions

Our study shows that the in vitro synthesis of melanin-like materials yields two fractions: a colorless, dispersible fraction and a dark-colored, precipitated fraction. The colorless dispersible fraction absorbs light in the UVA and UVB range and has fluorescent properties. Our studies suggest that some of the properties ascribed to melanin may reside in the colorless fraction. It remains to be seen if our observations have any relevance to the functions and properties of melanin in vivo.

## Figures and Tables

**Figure 1 ijms-25-08490-f001:**
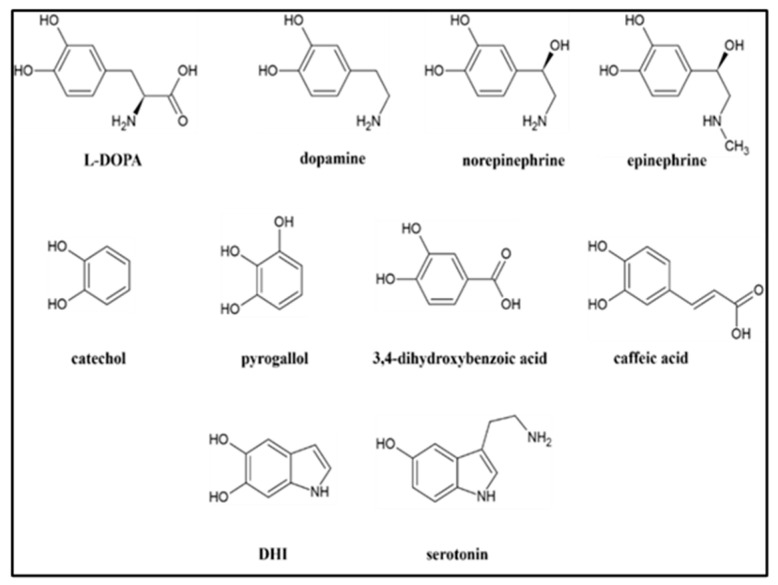
Chemical structures of the precursors used in the synthesis of MN-like materials.

**Figure 2 ijms-25-08490-f002:**
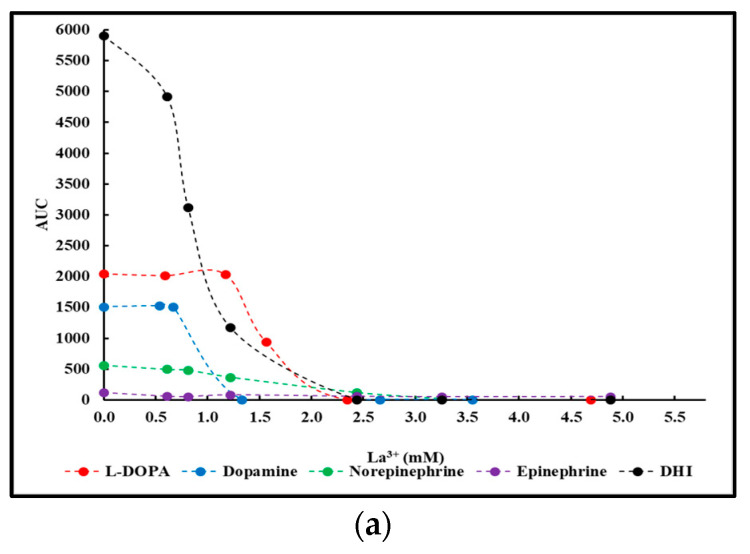
Changes in AUC values of dialyzed reaction mixtures following precipitation with varying amounts of La^3+^. (**a**) Reactions involving L-DOPA, dopamine, norepinephrine, epinephrine, and DHI. (**b**) Reactions involving catechol, pyrogallol, 3,4-dihydroxybenzoic acid, and caffeic acid.

**Figure 3 ijms-25-08490-f003:**
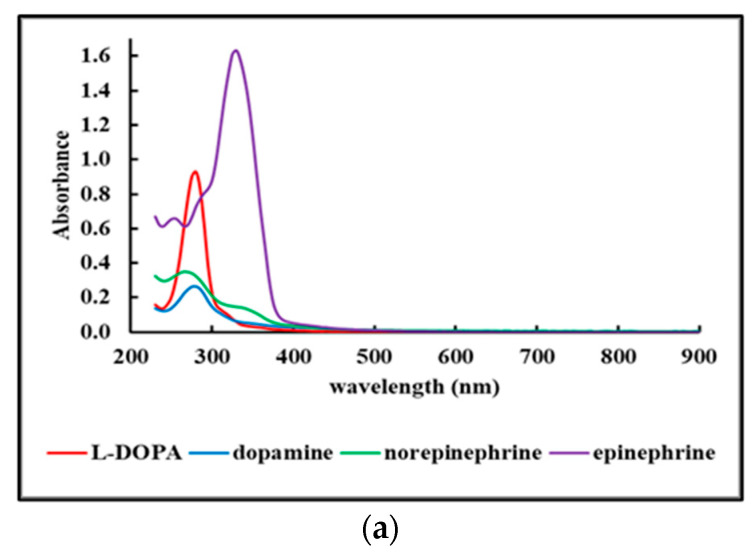
UV-Vis profiles of the F_disp_ fractions obtained from select precursors used in this study. The fractions were dispersed in water at a concentration of 100 μg/mL. (**a**) Reactions involving L-DOPA, dopamine, norepinephrine, epinephrine. (**b**) Reactions involving catechol, pyrogallol, 3,4-dihydroxybenzoic acid, and caffeic acid.

**Figure 4 ijms-25-08490-f004:**
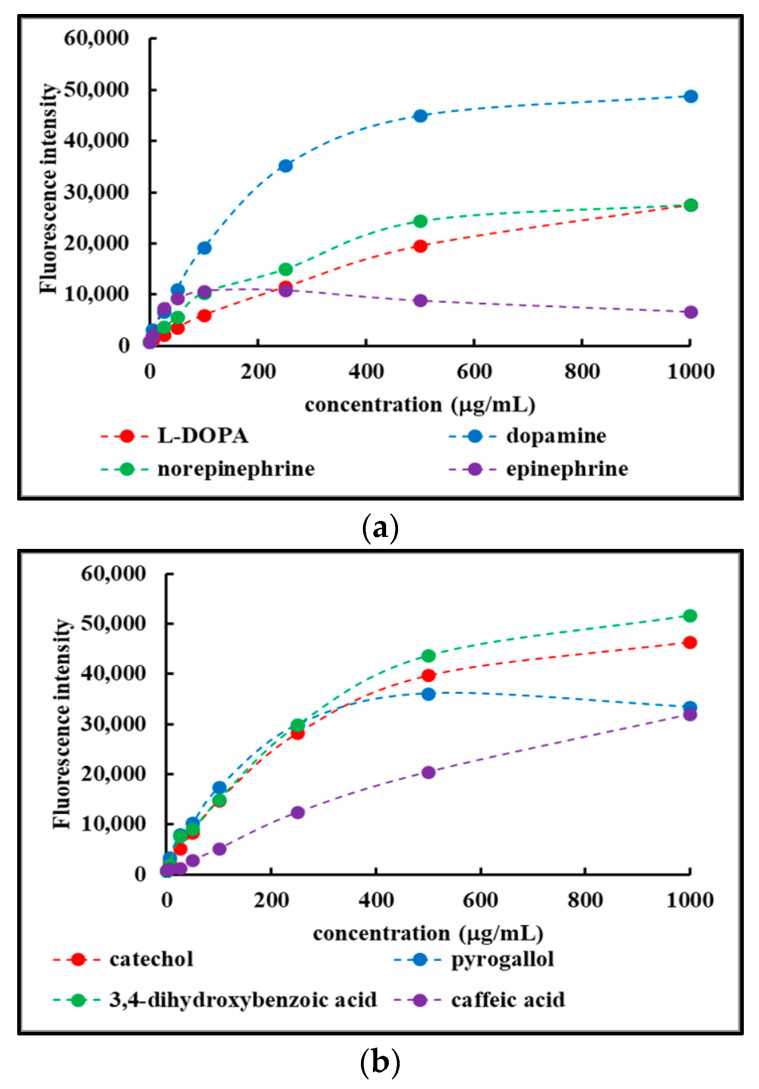
Concentration-dependent fluorescence of the F_disp_ fractions obtained from select precursors used in this study. (**a**) Reactions involving L-DOPA, dopamine, norepinephrine, epinephrine. (**b**) Reactions involving catechol, pyrogallol, 3,4-dihydroxybenzoic acid, and caffeic acid.

**Figure 5 ijms-25-08490-f005:**
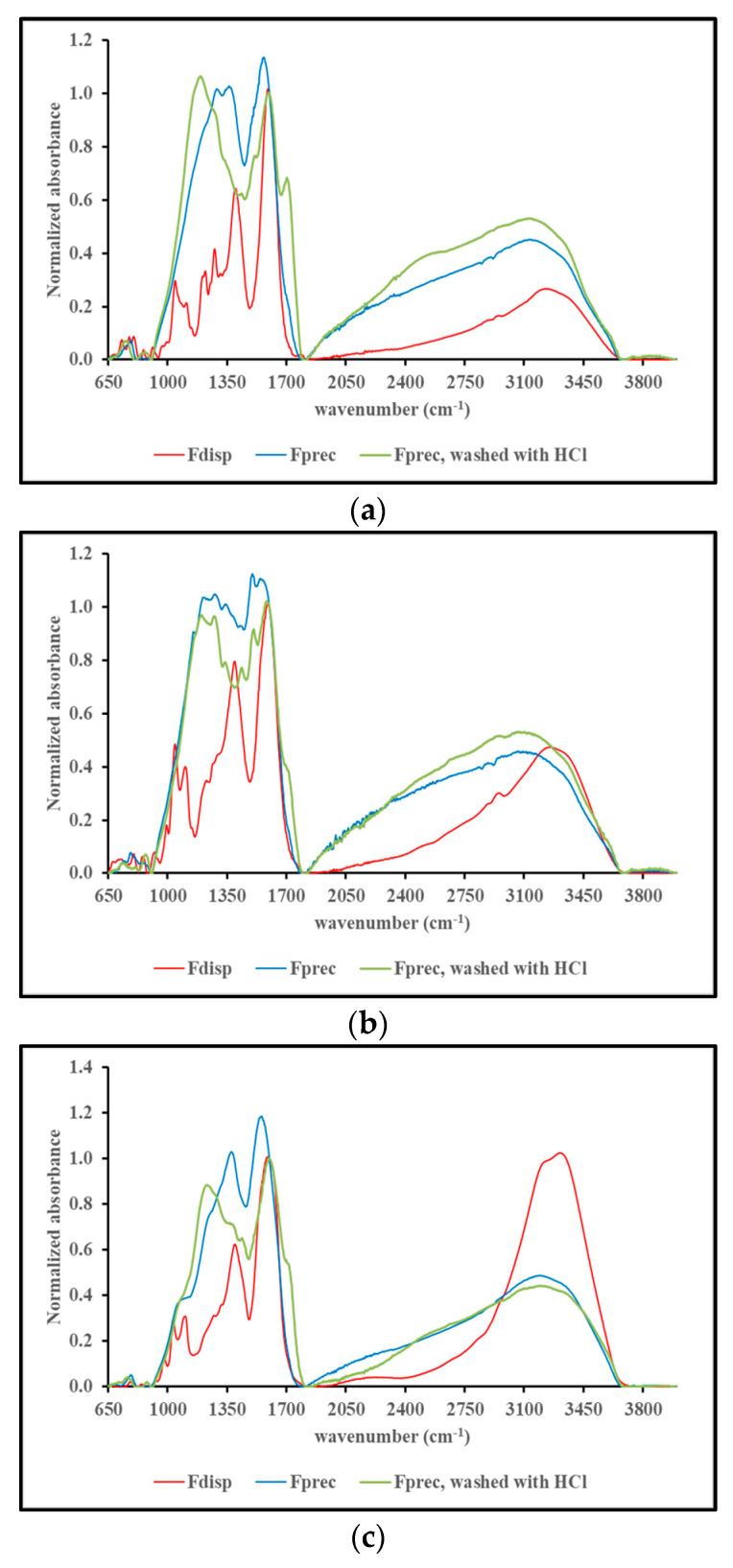
FT-IR spectra of the F_disp_, F_prec_, and HCl-washed F_prec_ fractions obtained from (**a**) L-DOPA, (**b**) dopamine, (**c**) norepinephrine, (**d**) epinephrine, (**e**) catechol, (**f**) pyrogallol, (**g**) 3,4-dihydroxybenzoic acid, (**h**) caffeic acid, (**i**) DHI, and (**j**) serotonin. For comparison purposes, all profiles are normalized for their absorbance at 1600 cm^−1^.

## Data Availability

The original contributions presented in the study are included in the article/Appendix A, further inquiries can be directed to the corresponding author.

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
