# Peer review of "The Invisible Fraction within Melanin Capable of Absorbing UV Light and with Fluorescent Properties: Is It Lacking Consideration?"

_ijms, 2024, doi:10.3390/ijms25158490_

Round 1

Reviewer 1 Report

Comments and Suggestions for Authors

Melanin is a pigment product of reactions from the melanogenic pathway. Melanin is important for the color of the eyes and skin. Genetic changes could cause the overproduction or the loss of melanin production. Melanin is poorly studied in terms of its chemical structure and some of its physical properties. In this work,  the molecular mechanisms of melanin formation are tested in vitro, Authors are using 10 different precursors to synthesize melanin-like materials, which were generated through air-oxidation in the presence of Na2CO3. After that, the melanin-like materials were fractionated into a dark substance and a near-colorless substance through the addition of LaCl3. The fractions were analyzed using spectroscopy methods. From this analysis, authors concluded that 'the in vitro synthesis of melanin-like materials yields two fractions: a colorless, dispersible fraction and a dark-colored precipitated fraction. The colorless dispersible fraction absorbs light in the UVA and UVB range and has fluorescent properties. The studies suggest that some of the properties ascribed to melanin may reside in the colorless fraction'.

The observation that after the air-oxidation-driven melanin synthesis, some of the melanin product is still in the form of colorless material is quite intriguing.

One of the possible questions is why melanin molecules are not forming large particles or precipitates. Unfortunately, this question does not have a clear explanation in the discussion. The finding of the colorless substance of melanin could be very important if the presence of the colorless substance is demonstrated in cell culture or in vivo. However, this finding is very interesting and could be important for future studies.

Reviewer 2 Report

Comments and Suggestions for Authors

The authors performed an experimental investigation on the optical absorption properties of melanin. They found that in vitro synthesized melanin can be fractionated into a dark-colored precipitate and a colorless, dispersible fraction. The colorless dispersible fraction absorbs light in the UVA and UVB range and has fluorescent properties. They conclude that their studies suggest that some of the properties ascribed to melanin may reside in the colorless fraction. They add that it remains to be seen if their observations have any relevance to the functions and properties of melanin in vivo.

-I believe this paper would benefit from several changes before it can be considered for publication.

-I find the title of their manuscript confusing. They should change it It is better to refer to the optical properties of melanin.

-In a precedent manuscript, one of the authors of the present manuscript refers to “invisible” ligands that stabilize colloidal melanin particles – the case of L-DOPA.  In that work, and based upon the results, a model for synthetic melanins as colloidal particles built from a dark-colored core aggregate stabilized by a set of colorless ligands is suggested. This conclusion changes the definition of “melanin,” known as the molecule C3H6N6. If some ligands are added to this molecule, its optical properties should change. I assume that the authors of the current manuscript are using their transformed definition of the molecule melanin. They must discuss this point in a clear way.

-In line 301, the authors mention, “The reality is that MNs often present themselves as physically stabilized, colloidal particles.” And they add: ”In an earlier report, focusing on L-DOPA, we have presented evidence that some colorless reaction products may serve as stabilizing ligands for the dark-colored MN particles.[34].” Do the authors consider the name “melanin” as the C3H6N6 molecule plus the “stabilizing ligands.” What are those ligands? They must make a discussion of this point.

-I cannot recommend the publication of this manuscript in its present form.

Round 2

Reviewer 2 Report

Comments and Suggestions for Authors

The authors investigated experimentally the invisible fraction within melanin capable of absorbing UV light and with fluorescent properties and asked if it is lacking consideration.

In my opinion, this paper requires changes before being considered for publication.

- I find the Introduction confusing. For example, the authors mentioned: “Melanin (MN), a ubiquitous class of darkly colored pigments, is poorly defined in terms of its chemical structure and some of its physical properties. Many reviews have been written about this enigmatic class of biomolecules detailing these unresolved issues.[1-8].”

-In the same section, they wrote that “a wide variety of such precursors can generate MN-like materials.”In the same section, they wrote, “Overall, the effect of MN, beneficial or detrimental, is uncertain. This uncertainty could in part be attributed to the uncertainties surrounding the chemical structure of MN. MN is not a single molecule but is typically considered a combination of chemically related entities. [10,19].” They also mentioned, “This report expands our earlier observations by expanding the pool of precursors used to synthesize MN-like materials.”

- For the readers’ benefit the authors must define the meaning of “MN-like materials”. Did the authors work only with “EuMN-like materials”?

-In the conclusions, the authors wrote ”Our studies suggest that some of the properties ascribed to melanin may reside in the colorless fraction.”

-If the “colorless fraction” is chemically integrated into the rest of the “Melanin-Like” material the optical properties of the whole are NOT the simple superposition of the molecules that integrate it. It is very well known that the doping of a material with atoms or molecules CHANGE its original optical and transport properties. Their conclusion is confusing. The authors must discuss this point to make it clear.

Round 3

Reviewer 2 Report

Comments and Suggestions for Authors

The authors made the requested changes.